# Assessment of Flood Forecast Products for a Coupled Tributary-Coastal Model

Robert Cifelli [1,*], Lynn E. Johnson [1,2], Jungho Kim [1,2], Tim Coleman [3], Greg Pratt [4], Liv Herdman [5], Rosanne Martyr-Koller [5], Juliette A. FinziHart [5], Li Erikson [5], Patrick Barnard [5] and Michael Anderson [6]

1   NOAA Physical Sciences Laboratory, 325 Broadway, Boulder, CO 80305, USA; lynnejohnson44@gmail.com (L.E.J.); jungho.kim@noaa.gov (J.K.)
2   Cooperative Institute for Research in the Atmosphere at the NOAA Physical Sciences Laboratory, Boulder, CO 80305, USA
3   Cooperative Institute for Research in Environmental Sciences at the NOAA Physical Sciences Laboratory, Boulder, CO 80305, USA; tim.coleman@nrel.gov
4   NOAA Global Systems Laboratory, 325 Broadway, Boulder, CO 80305, USA; greg.pratt@noaa.gov
5   USGS Pacific Coastal and Marine Science Center, 2885 Mission St., Santa Cruz, CA 95060, USA; lherdman@usgs.gov (L.H.); rosanne.martyr-koller@climateanalytics.org (R.M.-K.); jfinzihart@gmail.com (J.A.F.); lerikson@usgs.gov (L.E.); pbarnard@usgs.gov (P.B.)
6   California Department of Water Resources, 3310 El Camino Avenue, Sacramento, CA 95821, USA; Michael.L.Anderson@water.ca.gov
*   Correspondence: rob.cifelli@noaa.gov

**Abstract:** Compound flooding, resulting from a combination of riverine and coastal processes, is a complex but important hazard to resolve along urbanized shorelines in the vicinity of river mouths. However, inland flooding models rarely consider oceanographic conditions, and vice versa for coastal flood models. Here, we describe the development of an operational, integrated coastal-watershed flooding model to address this issue of compound flooding in a highly urbanized estuarine environment, San Francisco Bay (CA, USA), where the surrounding communities are susceptible to flooding along the bay shoreline and inland rivers and creeks that drain to the bay. The integrated tributary-coastal forecast model (Hydro-Coastal Storm Modeling System, or Hydro-CoSMoS) was developed to provide water managers and other users with flood forecast information beyond what is currently available. Results presented here are focused on the interaction of the Napa River watershed and the San Pablo Bay at the northern end of San Francisco Bay. This paper describes the modeling setup, the scenario used in a tabletop exercise (TTE), and the assessment of the various flood forecast information products. Hydro-CoSMoS successfully demonstrated the capability to provide watershed and coastal flood information at scales and locations where no such information is currently available and was also successful in showing how tributary flows could be used to inform the coastal storm model during a flooding scenario. The TTE provided valuable feedback on how to guide continued model development and to inform what model outputs and formats are most useful to end-users.

**Keywords:** coastal flooding; flood forecast; tributary-coastal; San Francisco Bay; tabletop exercise

## 1. Introduction

The San Francisco Bay (the Bay) area is home to over 7 million people and supports one of the most prosperous economies in the U.S. [1]. The area is highly urbanized, encompassing 9 counties, 3 major cities (San Francisco, Oakland, and San Jose), and US Highway 101 adjacent communities. The region is susceptible to coastal flooding along the bay shoreline as well as storm water and flash flooding along inland rivers and creeks that drain 483 watersheds [2] to the Bay (Figure 1). According to a 2013 California Department of Water Resources (CA-DWR) report, over 355,000 people in the Bay area are exposed in the 100-year flood plain and this number increases to over 1 million people

in the 500-year flood plain [3]. Estimated value of exposed structures in the 100- and 500-year floodplains are $46.2 billion and $133.8 billion, respectively. Water, transportation, and emergency management agencies across the 9 county San Francisco Bay region rely on flood forecasts to assess risk and inform their flood mitigation processes. Of special concern is combined coastal and tributary flooding events that compound the effects of either one alone (i.e., compound flooding events; [4]). This has been assessed by the U.S. Geological Survey (USGS) Coastal Storm Modeling System (CoSMoS) for the top 12 tributaries in a simplified manner (i.e., one-way coupling, daily flows; [5]), but not with real-time atmospheric forcing and coupling, or short time steps.

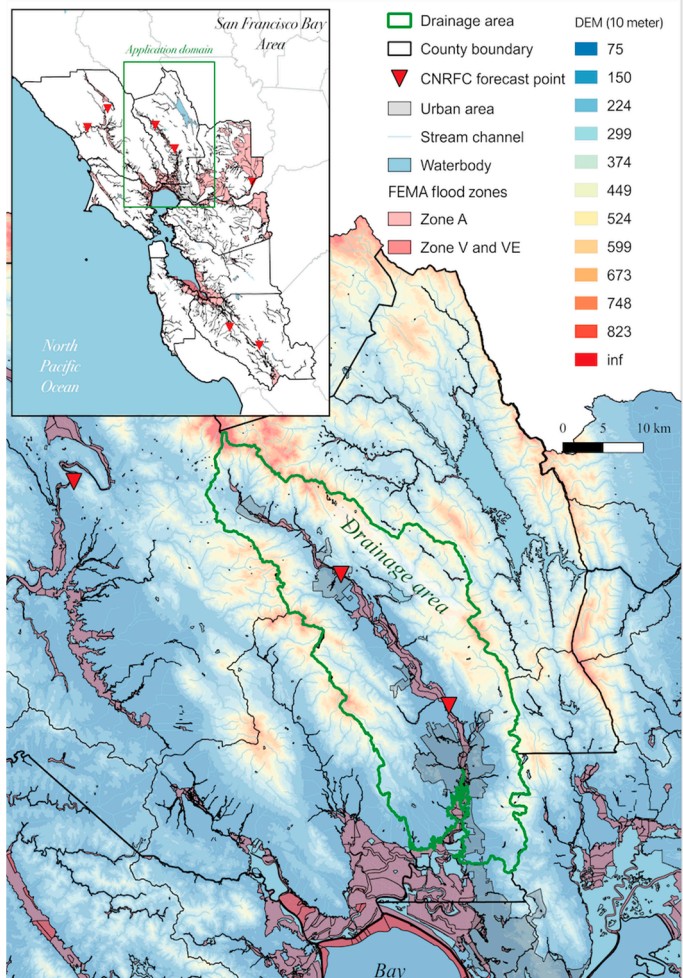

**Figure 1.** Map showing important geographic features in the region around San Pablo Bay at the northern end of San Francisco Bay where the case study described in the text was conducted. Insert shows the location relative to the greater San Francisco Bay area. The legend indicates color shading for elevation (m), Federal Emergency Mangement Agency (FEMA) flood zones A (Areas with a 1% annual chance of flooding), V and VE (Coastal areas with a 1% or greater chance of flooding and an additional hazard associated with storm waves), urban regions, and water bodies. The outline of the Napa river watershed, stream channels, and county boundaries are indicated by line colors corresponding to the legend. CNRFC forecast points are identified by inverted red triangles.

Currently, the National Oceanic and Atmospheric Administration (NOAA) National Weather Service (NWS) and the National Ocean Service (NOS) provide flood and high-water forecasts at selected points along major rivers surrounding the Bay and in and along the Bay. The California-Nevada River Forecast Center (CNRFC, https://www.cnrfc.noaa.gov/) has 22 river forecasts points on major inflows in the San Francisco Bay Area,

which are all upstream of the head of tide locations such that there is no accounting of the coastal influence on forecast water levels (see Figure 1). Site-specific forecasts for small tributaries surrounding the Bay are not provided except in a categorical manner (e.g., Flash Flood Guidance in [6]). In contrast, a distributed hydrologic model (DHM) can provide flow forecasts for every grid or stream reach in a watershed. For example, the National Water Model (NWM — see [7] for a description of the NWM) has more than 11,000 stream reaches in the nine county region surrounding San Francisco Bay. The NOS San Francisco Bay Operational Forecast System (SFBOFS, https://tidesandcurrents.noaa.gov/ofs/sfbofs/sfbofs_info.html) also provides water level forecasts in the Bay and the intertidal zone of the major tributaries. However, these coastal water level forecasts are mainly for navigation purposes and have relatively coarse resolution for estuary flood inundation. In the Bay area, the intertidal and small stream interface zone represents a significant fraction of the total area susceptible to flooding.

San Francisco Bay agencies responsible for flood mitigation, water supply, water quality impacts, and coordinating emergency response efforts have expressed a need for more accurate and timely information on precipitation, tributary flows, and coastal flooding in order to carry out their respective missions. To address the needs of water agencies for more information in the tributary and intertidal zone for both planning and real time operation purposes, the CA-DWR supported a joint NOAA-U.S. Geological Survey (USGS) project to develop a model prototype integrating a NOAA distributed watershed model and USGS coastal hydrodynamic model for the Napa River and estuary portion of north San Francisco Bay. The combined modeling system is referred to as Hydro-CoSMoS. The model prototype is viewed as a complement to statistical approaches, aimed at characterizing hazards associated with compound riverine and coastal flooding (e.g., [8]).

A flood event scenario was generated using the Hydro-CoSMoS model and demonstrated to water agency representatives through a tabletop exercise (TTE). An earlier project which focused on the DHM for the Russian-Napa rivers [9] involved an Advisory Panel of NWS and regional water managers to assess the usefulness of the DHM for decision making during heavy precipitation events. Agency representatives reviewed the forecast products and provided feedback on how to improve the system to support future real-time flood response operations.

The prototypes were initial steps as part of a larger follow-on effort to provide improved real time forecasts of precipitation, streamflow, and coastal flooding in the San Francisco Bay area. This effort, known as the Advanced Quantitative Precipitation Information (AQPI) system is being implemented in the 9 counties that border San Francisco Bay. Design and implementation of the AQPI system is guided by a risk reduction systems engineering process directed to ensuring that the system will be usable and acceptable to the various agencies and user communities. The process is illustrated as the "spiral model" (Figure 2). The process involves iterative development of system prototypes with assessment feedback to guide subsequent advancements. It begins at the center position and moves clockwise in traversals. Each traversal of the spiral results in a deliverable to be assessed. The first traversals may result in a requirements specification. The second will result in a prototype, and the next one will result in another prototype or sample of a product, until the last traversal leads to the final system. The figure quadrants illustrate tasks for (1) determining objectives, (2) identifying and resolving risks, (3) development and testing, and (4) planning for the next iteration. Of import for the AQPI system is the number and variety of agencies anticipated to use the system.

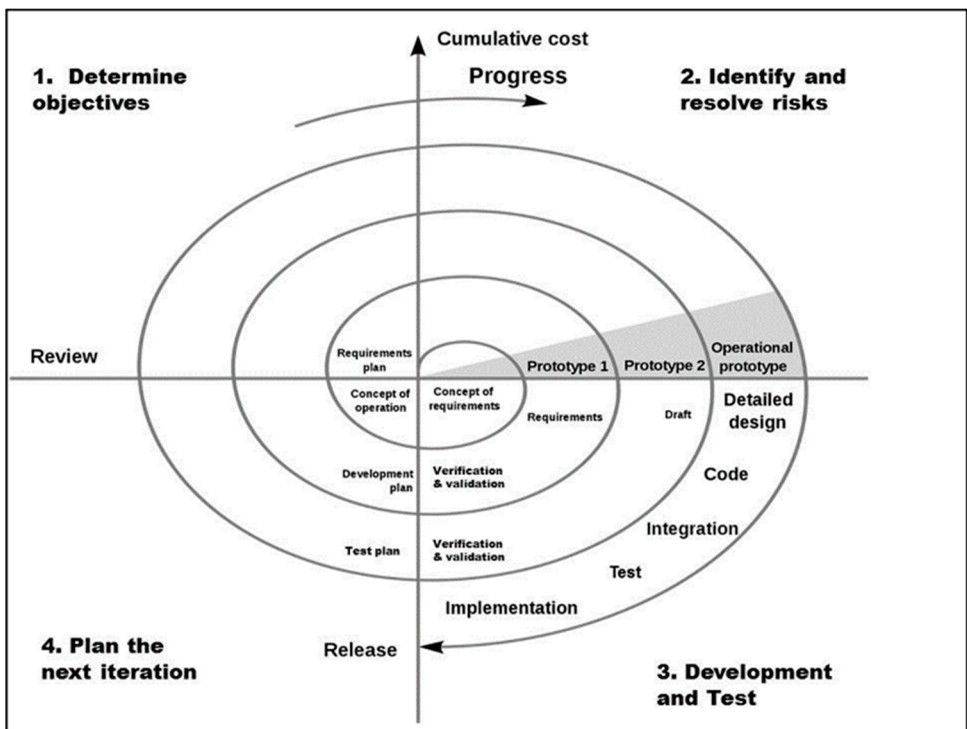

**Figure 2.** Spiral model of system development is intended to reduce risks that system will not meet user needs (after [10]).

Identification and resolving risks that the system will not meet user needs involves both technical and administrative dimensions. On the technical side the elements of a system involve major components for (1) data, (2) models, and (3) dialog or interface (e.g., [11]). For the AQPI system, the data and models are described in detail below, along with candidate displays for the interface as part of the assessments. Administrative aspects involve integration of the AQPI into flood threat assessment and warning procedures of the various agencies and other users. For example, many of the flood mitigation agencies (e.g., county public works departments) have developed their own capabilities for data collection and collation, and flood threat tools and models. Feedback obtained by AQPI prototyping activities described herein strongly advised that we integrate AQPI data and model outputs into the agency procedures.

Pertaining to the feedback and assessment methods, the study described herein builds on previous research aimed at development of forecast prototypes and associated risk-reduction programs. Examples include [12–15]. A menu of assessment methods includes questionnaires, product usage logs, evaluation logs, structured tasks, interviews, and tabletop exercises. Some aspects of respondent feedback and engagement can be captured using computerized tools (e.g., on-line questionnaire). Design of questionnaires and other feedback tools has important pedagogical aspects. Organizing for advisory panels, user groups and related outreach efforts provides continuity of communications. There are trade-offs among objectivity, cost to the evaluator, and cost to the respondent. An ideal method is one that is high in objectivity and low in cost to both the evaluator and respondent. None of the listed methods are ideal across the board. In general, more objective data (e.g., statistical verifications) are desirable, but usability has many social science factors that are not easily quantified. In many cases, subjective anecdotal information is useful, especially in learning of agency-specific flood warning tools and procedures. Most often multiple methods are applied to the same issue to obtain a more complete picture. Importantly, assessment program design is required that supports two-way communication from system developers to users, and from users to developers.

This paper describes the tributary DHM and coastal hydrodynamic model (CHM) and coupling of the watershed and coastal models in the Napa River watershed and estuary.

The CHM allows flood depths to be predicted by the hydraulic influence of both tides and watershed driven flows. The paper also describes the TTE and advisory panel processes to assess the utility of the watershed and coastal flood prediction systems and engage the potential users of Hydro-CoSMoS for their operations.

## 2. Integrated Flood Forecast System: Hydro-CoSMoS

The Hydro-CoSMoS modeling system consists of the NOAA Research Distributed Hydrologic Model (RDHM) for the fluvial component and the USGS CoSMoS for the coastal component [5,16–18]. Details on the individual components and the model coupling are described below.

### 2.1. Research Distributed Hydrologic Model (RDHM)

The NWS-Office of Hydrologic Development (OHD, now the Office of Water Prediction—OWP) RDHM [19–22] was used to simulate the tributary flows and the overall movement of water through the watershed using a nominal grid resolution of 4 km. Advantages of the distributed model over a more traditional lumped hydrologic modeling approach are associated with the spatial detail of flow predictions at any grid location throughout the basin. RDHM represents the general functionality of the class of DHMs operating on a gridded data structure.

During the lifetime of this project, the NOAA NWM was introduced as the new DHM for the NOAA National Weather Service. The NWM is a higher resolution distributed model at 1 km compared to RDHM (see description of the NWM here — https://water. noaa.gov/about/nwm). It is anticipated that the NWM will be used in lieu of RDHM in the future for the AQPI real-time system development.

The RDHM is a conceptual hydrologic prediction model that can be used to account for runoff, streamflow, soil moisture, snowmelt, evapotranspiration, and various hydrologic states during storm events and inter-storm periods (Figure 3). The required inputs are precipitation and temperature. RDHM computes the water balance between precipitation and infiltration for each grid, and routes both surface and subsurface water flow based on conceptual representations of terrain, soils, vegetation, and the influences of these on infiltration and evapotranspiration.

A real-time forecasting prototype of RDHM in the Russian and Napa watersheds was established and operated at the NOAA Earth System Research Laboratory (ESRL) during the development of Hydro-CoSMoS. The Office of Water Prediction provided base data sets on terrain and channel networks, soils, and the default parameters for the RDHM model. The CNRFC provided the primary datasets on precipitation and temperature fields. A Hydromet Visualization Tool (HVT — described in more detail in Section 4 and [9]) was developed to provide an opportunity for users to interact and provide feedback on model product visualizations. The real-time system was run for two winter storm seasons and forecasts were run out to 48 h. For the Napa River basin, the simulations involved forecasted precipitation and surface runoff computations for each grid, and routing of the surface flows to the basin outlet. The grid flows could be visualized in the HVT as the flood recurrence interval equivalent (e.g., 100-year flood level), and on-line users could click on a grid to obtain the forecast runoff hydrograph. Flood impact features, such as road-stream crossings, could also be identified as a warning aid for emergency responders. At present, neither the RDHM nor the NWM are able to represent inundation of flood plains.

For the coupled Hydro-CoSMoS scenario additional precipitation data was derived from the Multi Radar–Multi Sensor (MRMS; [23,24]), an operational system that provides a suite of gridded quantitative precipitation estimation (QPE) products at 1 km spatial and hourly temporal resolution. The RDHM was applied for a retrospective run from 1 October 2010 to 31 March 2012, where the period prior to 2012 was used for warm-up and calibration purposes and January 2012–March 2012 was used for verification.

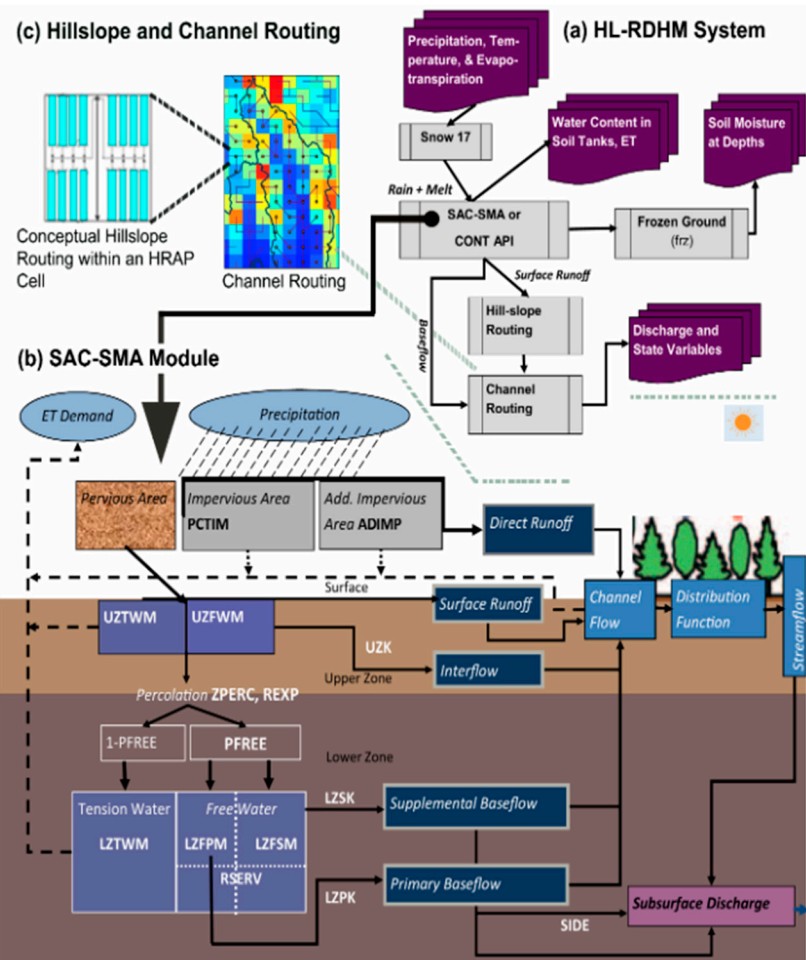

**Figure 3.** Components of the RDHM.

## 2.2. Coastal Storm Modeling System (CoSMoS)

The Coastal Storm Modeling System (CoSMoS) is a dynamic modeling approach that has been developed by the USGS to allow more detailed predictions of coastal flooding due to tides, storm surge, winds and tributary inflows [5,16–18]. It has also been applied for both future sea level rise and storms integrated with long-term coastal evolution (i.e., beach changes and cliff or bluff retreat) over large geographic areas (100 s of kilometers) along the west coast of the U.S., including San Francisco Bay. In general, CoSMoS is a framework that takes large scale oceanic conditions and scales them using regional and local models to generate high resolution hazard predictions ([16], Figure 4).

The implementation of CoSMoS used in this study is described in detail in [25]. Briefly, for the coastal application Delft3D-FM [26–28], a hydrodynamic model based on a flexible mesh grid, and Simulating Waves Nearshore (SWAN) [20], were used to calculate water levels that account for tidal forcing, seasonal water level anomalies, storm surge, and in-Bay waves derived from the wind and pressure fields of a NWS forecast model (North American Mesoscale Forecast System-NAM, see https://www.ncdc.noaa.gov/data-access/model-data/model-datasets/north-american-mesoscale-forecast-system-nam).

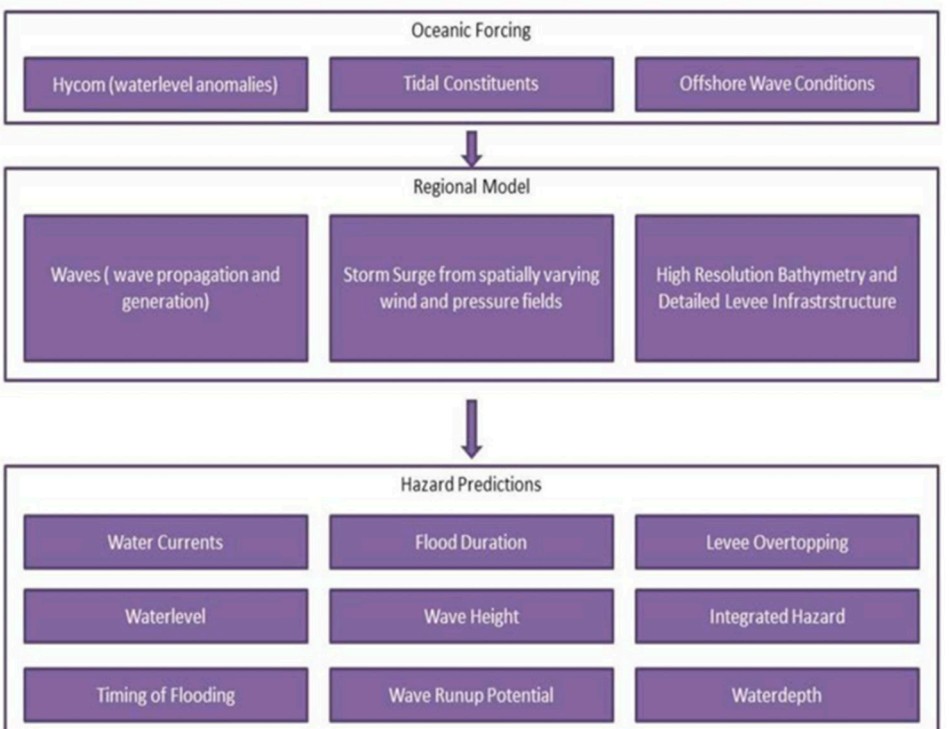

**Figure 4.** CoSMoS components.

Napa River discharge was retrieved from the RDHM as a time series just above the head of tide. The flooding extent using CoSMoS was determined by overlaying the resulting maximum water levels onto a 2-m digital elevation model of the estuary that resolves the extensive levees and tidal marshes. The performance of the model was evaluated based on comparisons with NOAA tide gauges around San Francisco Bay for two coastal flooding events. The evaluation showed that the model performed well at all locations and was consistently better than predictions using only tidal forcing [25,29].

## 3. Case Study Scenario: Napa River Basin and Estuary

### 3.1. Overview

The Napa River watershed is 1100 km$^2$ (426 mi$^2$) and is a mix of urban (~10%), agricultural (~35%), grassland (~15%), and forests (~40%), with the area adjacent to San Francisco Bay currently being restored to wetland habitat. It extends from the Mayacamas Mountains to the north and empties into San Pablo Bay at the north end of San Francisco Bay, west of the Carquinez Strait (Figure 1). The watershed is bounded by relatively steep terrain surrounding the long narrow valley that is 43 km long (28 mi) and 8 km (5 mi) wide at its widest point [30]. The City of Calistoga is in the northern end of the watershed and the City of Napa sits at the southern, tidally influenced end, with Vallejo, CA located on the eastern side of the river where it meets the Bay (Figure 1).

The Napa River and estuary were chosen for the prototype development and assessment because of the history of flooding and the influence of tides up to the City of Napa. The Napa River has a long history of flooding; since settlers began keeping track of such notable events, 21 serious floods have been recorded from 1862 to the present day (https://www.countyofnapa.org/1094/The-History-of-Floods). According to the U.S. Army Corps of Engineers, a major event in 1986 flooded the City of Napa, requiring the evacuation of over 5000 people and producing $100 million in damages (https://www.spk.usace.army.mil/Missions/Civil-Works/Napa/). This flooding history prompted development of a flood bypass channel in the City of Napa, details of which were included in the CoSMoS model.

A realistic storm scenario for the Napa River watershed and estuary was created to demonstrate the use of Hydro-CoSMoS to emergency responders and planners. Like the ARkstorm scenario [31], where a hypothetical storm was constructed to represent a plausible series of precipitation and flooding events, the Napa River-Estuary scenario was based on a combination of actual events that have occurred in the San Francisco Bay area and consisted of two components: the watershed and the estuary. Although the actual events did not occur together, they represent a plausible combined flooding event, based on the historical record as noted above. For the watershed, soil moisture, and precipitation conditions were scaled up (as described below) and input into the RDHM, providing projections of fluvial-related flooding. For the estuary, flooding was projected from a scaled-up event with waves, winds and atmospheric pressure modeled using CoSMoS in San Francisco Bay. The storm scenario combined these two components over a 3-day (72 h) forecast period to provide projections for the watershed and coastal area in the Napa River basin and estuary. The 3-day period was selected based on personal communication with flood managers in the bay area as a time period during which they could reasonably prepare for a storm. Details of the storm scenarios are provided below.

### 3.2. Watershed Scenario

An event that occurred over Napa County on 23 December 2012 was selected as a basis for the watershed storm scenario. This storm was typical of rain events that occur in the Napa region during the October–March wet season, with the greatest amount of rainfall in the higher elevations of the upper watershed and lesser amounts closer to the Bay. The 23 December event produced 58.3 mm (2.29 in) of rainfall averaged over the Napa watershed corresponding to a 1-to-2-year return period for 12-h precipitation. However, the CNRFC issued a flood warning because of the high peak flow at the USGS gauge site 11,458,000 located near the city of Napa, CA (371 cms (13,100 cfs) at 08:00 PST, December 24, 2012) corresponding to a 5-year return period for streamflow. The prototype RDHM simulated a peak flow of about 482 cms (17,000 cfs). In addition, soil moisture conditions, based on NOAA soil moisture observations in the nearby Russian River watershed, indicated 46% saturation at 10 cm depth, much higher than the normal saturation (23% at 10 cm depth; [32]). This storm was then scaled-up to reflect a more significant flooding event, as described next.

To scale this storm to a more extreme event for the purposes of the scenario, the rainfall and soil moisture conditions were modified. The 25-year recurrence interval rainfall fields for the Napa watershed were estimated through NOAA Atlas 14 [33] and then merged with the actual MRMS radar rainfall data. This process increased the overall magnitude of the event while preserving the spatial-temporal characteristics of the actual 2012 storm event [32]. The maximum soil moisture condition was also increased based on an actual storm event that happened on 27 March 2012. The resulting precipitation and soil moisture conditions were input into the RDHM and the river basin streamflows were simulated (Figure 5). The basin outflows were then input to the CoSMoS model which simulated estuary flood inundation depths (Figure 6). Additional details on the watershed scenario are provided in [25,32].

### 3.3. Coastal Scenario

For the coastal scenario, a storm was chosen that, while extreme, represented a reasonable and plausible event. The atmospheric and wave conditions were chosen to match historical conditions that created an observed 50-year return period non-tidal water level at the San Francisco gage station located on the south-east side of the Golden Gate Bridge. Water level data were analyzed from 1890 to present-day to determine the non-tidal residuals and the 50-year return period non-tidal water level was found to be near 44 cm (1.44 ft). Non-tidal water levels reached a peak of 44.1 cm at 4 pm on 6 February 1998. The wind, barometric pressure and wave conditions from that time were taken as representative conditions for a 50-year storm.

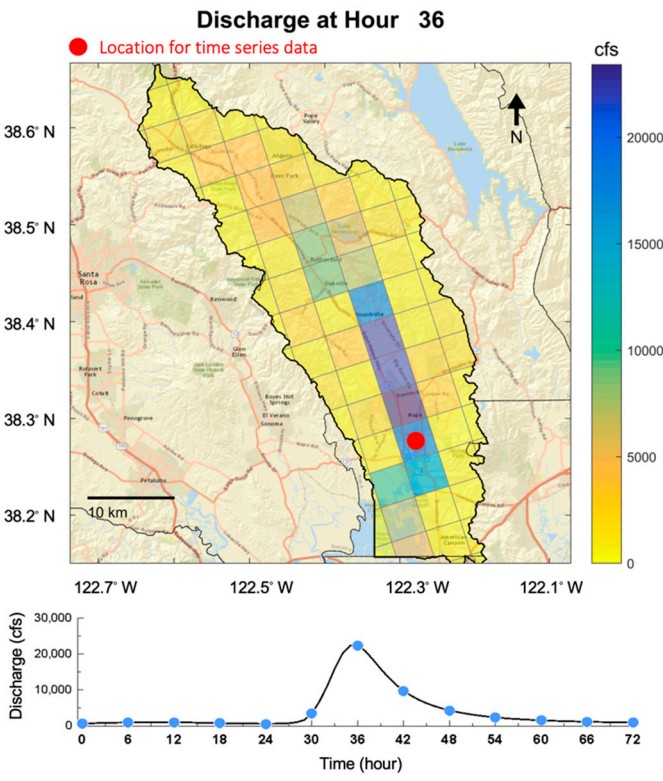

**Figure 5.** Output from the watershed component of Hydro-CoSMoS showing (**top**) discharge across the Napa River watershed at hour 36 during the storm scenario and (**bottom**) the basin outlet streamflow hydrograph generated from the watershed scenario that was used as input into CoSMoS. The location of the hydrograph is indicated by the red dot.

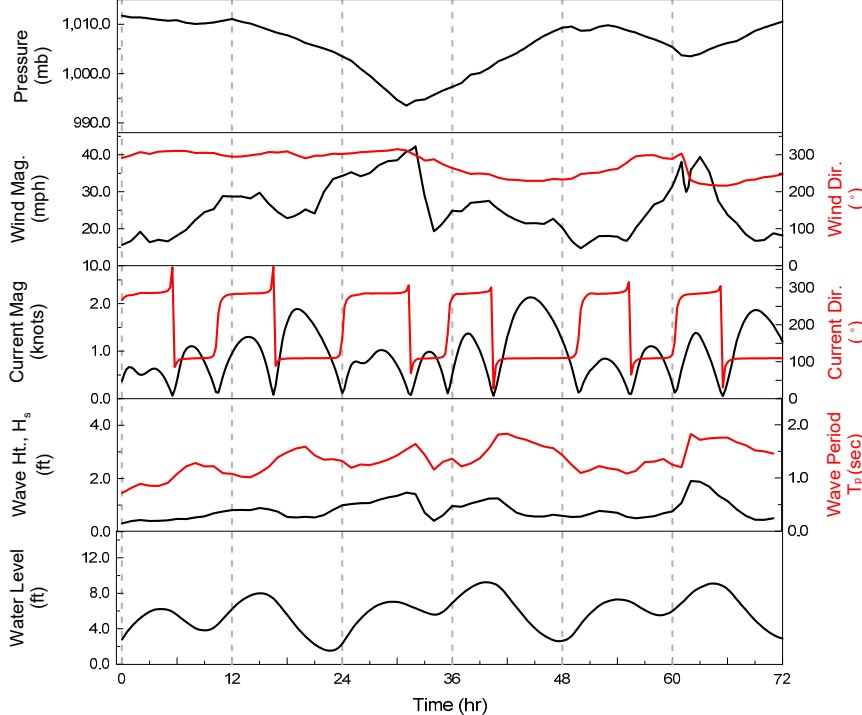

**Figure 6.** Time series output from the coastal component of Hydro-CoSMoS at the Napa River mouth for the storm scenario described in the text (from [25]).

During this 50-year storm wind speeds ranged from 4 to 12 m/s (9 to 27 mph) and were predominantly from the south-west direction. The atmospheric pressure reached a low of 984 mb in this storm. Offshore waves in the Pacific ranged from 3.4 to 8.2 m (11.2 to 26.9 ft) significant wave height and had a peak period of 20 s with most of the energy coming from the west. These atmospheric storm and wave conditions were applied to a spring tidal period from November 2010. These tides represented slightly higher high tides and slightly lower low tides than average (i.e., a larger tidal range) but they are not as significant as king tides which occur later in the winter.

Additional effort was directed to defining the bathymetry of the Napa River near the City of Napa, as there has been recent construction of a flood bypass channel (https://www.spk.usace.army.mil/Missions/Civil-Works/Napa/). A 2-m digital elevation model was obtained and used to define the estuary and river channel up to and upstream of the City of Napa. Preliminary simulations indicated that tidal and storm surge influences would be felt at this location and further upstream. Field investigation provided details on the bypass channel configuration.

*3.4. Coupling the Watershed and Coastal Models*

The NWS Flood Early Warning System (FEWS) provided the general framework for the coupling of the RDHM and CoSMoS models into Hydro-CoSMoS. FEWS is an open shell system for managing input and output communications that is extensively used in NWS forecasting products and for handling real-time time-series data (http://oss.deltares.nl/web/delft-fews/). FEWS was also used for the earlier RDHM prototype for the Russian-Napa Rivers. Although the model coupling could have been done outside of FEWS, this framework was used as a template that could be copied for a future real-time application of Hydro-CoSMoS. The coupling process involved retrieval of the weather forecast information required to run CoSMoS as defined above and provide the inputs for RDHM. The hydrological model was run to provide a time series of discharge for the Napa River which was then passed to CoSMoS to make water level predictions (Figures 5 and 6). Note that Hydro-CoSMoS did not involve full two-way coupling between RDHM and CoSMoS. A goal in this study was to demonstrate the feasibility of linking the models through the exchange of data in order to reveal compound effects of tributary and coastal flooding for stakeholder feedback. Future implementations as part of the AQPI project will incorporate more sophisticated and real-time coupling between the watershed and coastal model components.

## 4. Prototype Hydro-CoSMoS Flood Products

A number of visualization tools and products were developed as part of the combined coastal and tributary flooding scenario to facilitate assessment in the TTE described below. On the watershed side, the HVT was developed to produce real-time web-oriented displays of DHM outputs as animations of precipitation, flood runoff, flow frequency [20], soil moisture, and ancillary GIS mappings of flood impact features (e.g., bridge crossings). The HVT used a Google Maps interface so that it could be widely deployed and accessed using a commonly available platform familiar to most users. An Advisory Panel comprised of federal, state, and local flood managers reviewed the HVT displays and provided feedback on the various products. Detailed descriptions of HVT functions and feedback are provided in [9] and the salient features are summarized here:

- The tool was made available online via the web, without requiring software downloads by the user. The tool displayed GIS layers along with DHM grid results in raster Keyhole Markup Language (KML) format overlaying a Google Maps view of the study region.
- RDHM data were automatically loaded into the interface in real-time as data were made available by FEWS. Data available in the HVT were obtained by import from the FEWS and included gridded precipitation, soil moisture, and surface flows, for the DHM domain (i.e., Russian-Napa Rivers) for each time step (4-km grid, 1-h) and

overlays of flood frequency levels (e.g., 20-year flood frequency level) were added (Figure 7) to help users assess the relative magnitude of a flow event and where flash flood emergency responders may be needed. This capability was provided for all streams, including small tributaries, which had no flood information or flow gaging instrumentation.

- Users could look at specific day and time combinations and interact with the DHM data for specific grids. Pop-ups provided for specific grid points allow users to view and interact with the data in graph (i.e., hydrograph) and tabular format.
- The HVT also displayed at-risk road crossings (Figure 8) and other flood impact features (e.g., schools and health care facilities) on user request.

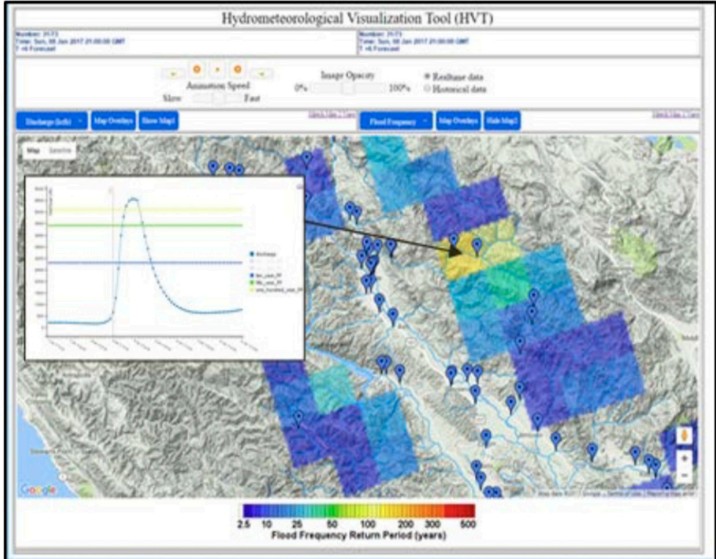

**Figure 7.** Example of HVT display showing a time series of streamflow (hydrograph) at a user selected location. The hydrograph indicates flood frequency levels associated with the grid element corresponding to the user-specified location.

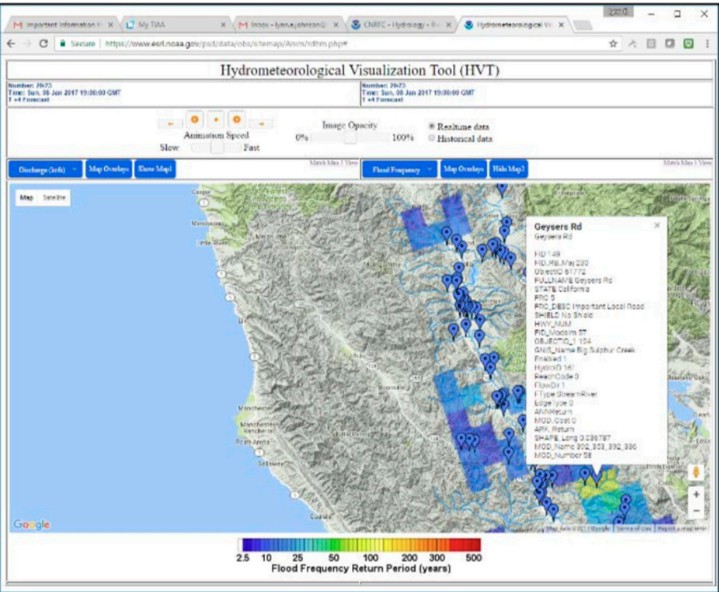

**Figure 8.** HVT display showing locations of road-stream intersections as a critical infrastructure to assess flooding risk.

For the coastal scenario, a variety of products were generated, including flood depths and extents, currents, water levels, timing and duration of flooding, wave heights and wave run-up potential, levee overtopping, and an integrated hazard metric. Flooding extents and depths (Figure 9) were determined by interpolating water levels (a direct output of the regional model) onto a 2 m resolution grid and subtracting the digital elevation model. Currents and wave heights were also direct outputs of the model. These results were output on hourly intervals, from which the spatial extent and the timing of initial flooding, the timing of the maximum flooding and the duration of the flooding could be determined. Given computational time constraints wave run-up could not be directly computed, but by using the detailed bathymetry and standard wave run-up formula from the engineering literature [34,35] estimates of wave runup could be made in the domain which can contribute to extended flooding. The water levels from this wave activity could be compared to the digitized levee network developed by the San Francisco Estuary Institute (SFEI, https://www.sfei.org/data/sf-bay-shore-inventory-gis-data#sthash.Zxnj0 UBX.dpbs) to predict potential levee overtopping. Finally, several integrated hazard metrics were computed that included time integration of currents and water depth to find the locations which will be most impacted in a storm event, either through very fast currents, extremely deep water or extremely long flood duration, or some combination of the three.

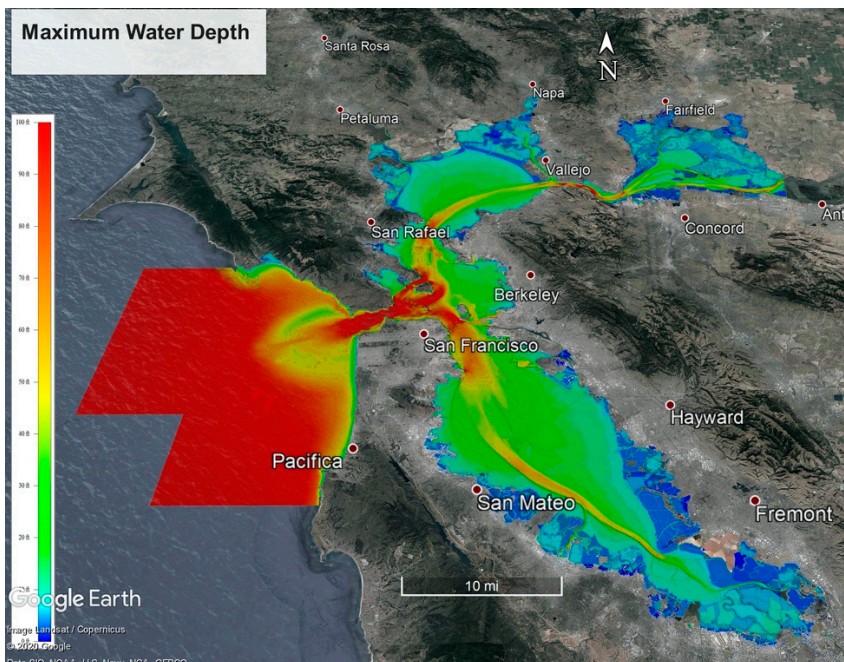

**Figure 9.** Hydro-CoSMoS display of simulated maximum water depth in San Francisco Bay for the scenario described in the text.

## 5. Table-Top Exercise (TTE)

A TTE was conducted to advance users' understanding of the Hydro-CoSMoS modeling system, provide feedback on the usefulness and usability of the various flood forecast products, and to inform the design of a fully operational watershed-coastal flood forecasting system for the AQPI project. TTEs are often used to facilitate discussion of a simulated emergency, in this case an event as defined by the watershed and coastal flood scenario described above. TTEs help increase understanding of technical details and information products, clarify roles and responsibilities, and identify additional mitigation and preparedness needs. These exercises typically result in action plans for continued improvement of the flood forecasting system. The Federal Emergency Management Agency (FEMA) has an Emergency Planning Exercises web page which provides guidance and resources for TTEs (https://www.fema.gov/emergency-planning-exercises). For this TTE, members of the

flood forecasting and emergency response community were asked to review and discuss the flood forecasting information and actions they would take, testing their understanding of the modeling outputs in an informal, low-stress environment.

A combined tributary and coastal flooding scenario was developed and used for the TTE as a venue to communicate the flood forecast model outputs to flood forecasters and emergency response managers. As noted above, the TTE was not intended to simulate an actual event; rather, a plausible scenario that could realistically occur. The TTE was intended to inform staff with the CA-DWR, NWS, and San Francisco Bay area counties about the capabilities of the coupled watershed and coastal flood prediction system, and to seek feedback on how forecast products could better serve their needs. The feedback will inform the design of the AQPI system for the entire 9 county SF Bay area.

The TTE was held at the CA DWR Flood Operations Center (FOC) in Sacramento, CA (Figure 10). The FOC is an advanced computing and networking facility used by the CA-DWR and NWS before and during flood events to assess risks, and coordinate flood response actions and communications with the various local, state, and federal agencies and citizens. The FOC has a collection of networked computers and teleconferencing equipment by which to display hydrometeorological forecasts and for interagency coordination. The exercise involved attendees from a variety of local, state, and federal agencies on-site, as well as webinar remote access for at-distance participants. Participants in the exercise included a collection of managers and staff of the CA-DWR, various county-level and regional flood agencies, California Department of Transportation and the NWS. A total of 12 participants attended on-site and an additional 14 joined on-line. A complete list of participants is shown in Table 1.

As described below, the TTE involved the interaction of users with products from both the scenario and an actual flooding event in 2017. To facilitate demonstration of the functionality of the Hydro-CoSMoS system, a "clickable PDF" document was created to provide an interactive medium for exploration of watershed and coastal model data and forecasts by the participants (Figure 11). The document was provided to participants beforehand and was presented and reviewed at the outset of the exercise sessions. Participants were also able to use desktop computers during the TTE to browse through the various flood model products in a manner similar to how they might be exposed to flood forecasts in real time. For instance, hydrological and meteorological data and flood forecast information products were time sequenced over the forecast period of 72 h as described in the storm scenario above.

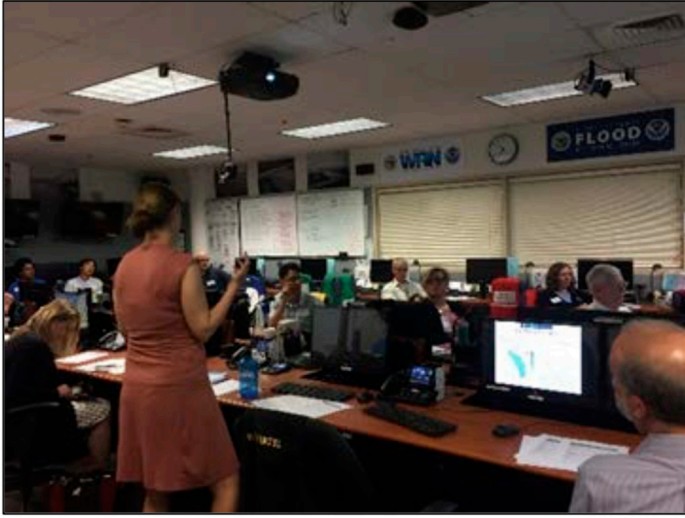

**Figure 10.** Photo captured during the TTE held at the CA-DWR Flood Operations Center demonstrating the Hydro-CosMoS.

**Table 1.** Agencies participating in the TTE.

| Agency | No. Participants (On-Site + On-Line) |
|---|---|
| Bay Area Flood Protection Agencies Association | 2 + 1 |
| California Department of Water Resources | 4 + 3 |
| California Department of Transportation | 1 + 3 |
| Contra Costa County Flood Control District | 0 + 1 |
| Marin County Public Works | 0 + 1 |
| Napa County Flood Control and Water Conservation District | 1 + 1 |
| National Weather Service California Nevada River Forecast Center | 1 + 1 |
| National Weather Service Monterey Weather Forecast Office | 0 + 2 |
| National Weather Service Sacramento Weather Forecast Office | 2 + 0 |
| University of California Davis | 1 + 1 |
| Total | 12 + 14 |
| US Geological Survey-Coastal Modeling Team | 3+ 2 |
| NOAA Earth System Research Lab | 3 + 1 |

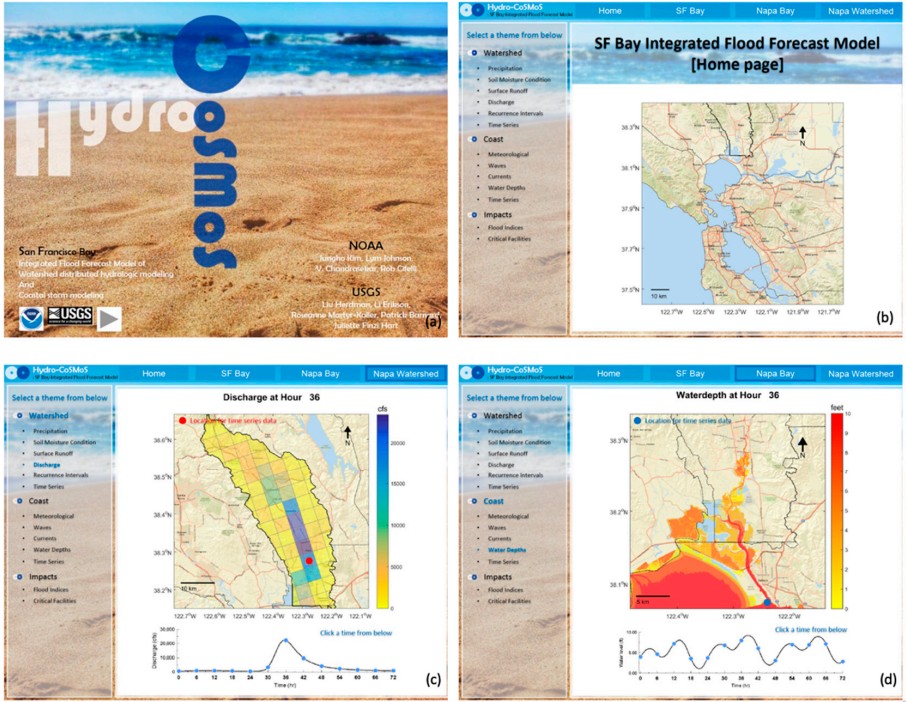

**Figure 11.** Example products available in the clickable PDF document used for the TTE. (**a**) document title page; (**b**) document description; (**c**) example discharge plot; and (**d**) example coastal water depth plot.

The flow of the TTE involved the following activities:

- Step 1: The Hydro-CoSMoS scenario (as described above) was outlined for the participants.
- Step 2: Participants in the room were divided into small groups. These groups and participants on the phone were given 15 min to review the flood scenario data via the PDF. A questionnaire was used to guide participants through review of the clickable PDF and provide initial responses to the modeling outputs developed for the scenario. In the questionnaire, respondents were asked information about their organization, their needs for coastal, estuarine and watershed flooding information, and what types of flooding information and products they currently use. They were then asked to review the clickable PDF and provide initial responses on the usefulness of the

various products (Figure 12). Following the small group exercise, participants were asked to provide their feedback in a larger group discussion. Notes were captured on flip charts.

- Step 3: In addition to the scenario, information was provided about actual flooding that had occurred during the winter of 2017 along a stretch of Route 37 in Napa County (Figure 13). Participants were then guided to assess the impacts from the scenario to a stretch of Route 37 in Napa County based on Hydro-CoSMoS modeling and outputs and relate that to the actual flooding information. The participants were asked to consider the following questions to determine the usefulness of the example products:

  1. Have you used information similar to this information before? If yes, what is the source?
  2. At what physical scale would you want this information?
  3. Are the time intervals of the forecast products appropriate?
  4. How else would you like to see this information displayed?
  5. Does the scale of the event matter? (i.e., Does a bigger event require different information than a smaller event)

**Figure 12.** User survey forms for the TTE. (**a**) Participants in the room were provided a hard copy printout in which they could indicate the usefulness of the model product. (**b**) Participants on the phone were provided a link to an online survey in which they could also provide real-time feedback.

During group discussion, representatives from Napa County provided insight into how the 2017 flooding was addressed, which information they used at the time, and how the information provided by the Hydro-CoSMoS model could enhance their response capabilities in the future.

- Step 4. The entire group discussed overall impressions of the modeling system and outputs and products, and identified next steps for the project. An assessment of the utility of Hydro-CoSMoS products and overall impressions from the TTE were developed from the questionnaire that was distributed to participants during the TTE.

Discussion throughout the TTE provided valuable information for the project partners. Prior to and during the exercise, participants were asked to rate the various model outputs as Not Useful, Somewhat Useful, or Very Useful. In general, most of the outputs presented were deemed as useful. For instance, for the watershed forecasts, time series outputs, precipitation information and flow recurrence were deemed to be Very Useful, while grid displays of runoff were rated as Not–Somewhat useful. Most of the proposed coastal flood forecast products were equally identified as Somewhat–Very Useful. Detailed results of

the survey are provided in the Appendix A and Tables 2 and 3 highlight some of the key findings in a tabular format to help synthesize the information received.

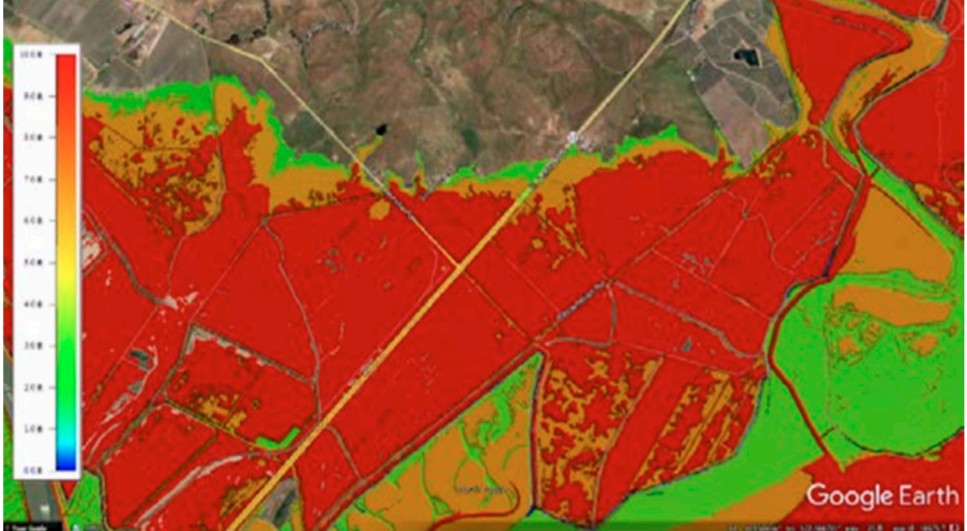

**Figure 13.** Visualization of flood potential over Highway 37 that was available for participants to explore during the TTE.

The highest rated products were the outputs that showed the extent, duration, and height of the flooding. Some users had fairly developed procedures for ingesting upstream data like precipitation forecasts that they then used to generate their own flood risk warnings. Given the local knowledge and sometimes high-resolution models that are utilized by these users, our regional scale model may not be able to outperform their predictions. However, many of the users engaged in this process also did not have sophisticated models or processes for forecasting extent, duration, and height of flooding. One big advantage of this system is the ability for users to tap into any of the levels of existing forecast that best matches their needs and capabilities. The feedback indicates it will complement the needs of more sophisticated users, and advance the capabilities of all the users in the region to understand flood risk.

Beyond the model outputs, TTE participants also discussed the most effective ways to contextualize the data outputs and how to make them the most useful to a wide range of audiences. For instance, the modeling team provided an initial list of critical assets and their exposure to the combined flooding; TTE participants agreed it would be beneficial to all audiences to expand the list of critical assets included in this analysis. There was also discussion of identifying the intended audience of the information. If these data were to be provided only to flood control and emergency response personnel, the information should be presented in formats they traditionally use and that the most critical questions to address are "Where, How High and When". However, if the intent is for this information to be more widely-accessible, there were numerous suggestions on how to make the information more relatable, such as: including geographic landmarks or known flood stage events on the maps so people could easily recognize their communities and incorporating the FEMA Floodplain maps as the floodplain delineations are commonly known and recognized.

**Table 2.** Summary of TTE questionnaire responses for tributary flooding information.

**Tabletop Exercise Questionnaire Responses**

| Question Topic | Subtopic | % Ratings by Category | | | Weighted Score | Comments |
|---|---|---|---|---|---|---|
| | | Not Useful 1 | Some what 3 | Very Useful 5 | | |
| Please describe your <u>job and needs</u> for flash flood forecasts. | Respondents included a mix of NWS forecasters, CaDWR flood managers, and several county-level flood response agency staff. A few are hydrologic modelers involved with R&D and local flash flood warning operations. | | | | | |
| Please describe your <u>needs for flood forecasts.</u> | NWS forecasters use radar data, flash flood guidance, rainfall reports, and river data for their warning decision making. County-level staff are concerned with storm surge into flood control channels and protecting communities and ecosystem restoration projects thereby. Some counties track flood threats and issue warnings. | | | | | |
| What <u>sources</u> do you currently typically consult for this type of information? | Most use NWS warnings, including from the WFO and the CNRFC. Some track precipitation reports (public and private), spotter reports (phone, email,, social media, and news media), USGS stream gages. Some county staff have their own precipitation and stream gage networks. | | | | | |
| Please provide your first-level reaction for the information provided in the WATERSHED section. Note that the WATERSHED data included DHM input factors that contribute to flood magnitude, such as precipitation (current and forecast). Saturated soil moisture levels can exacerbate flood runoff. Surface runoff is the local runoff from a single grid, Discharge is the cumulative runoff at a grid from all upstream grids. Time series is the flow hydrograph which forecasts the peak flow and time-to-peak, and the duration of high water. | Precipitation | 0% | 40% | 60% | 4.2 | Precipitation information rated Very Useful. |
| | Soil Moisture Content | 0% | 80% | 20% | 3.4 | Soil moisture rated Somewhat Useful. |
| | Surface Runoff | 40% | 40% | 20% | 2.6 | Grid runoff rated Not Useful to Somewhat Useful. |
| | Discharge | 0% | 40% | 60% | 4.2 | Cumulative discharge rated Useful. |
| | Recurrence Intervals | 20% | 20% | 60% | 3.8 | Recurrence interval rated Very Useful by most, but some rated is Not Useful. |
| | Time Series | 0% | 0% | 100% | 5.0 | Time series unanimous Very Useful. |
| Additional comments on the WATERSHED section. | Use local clock time for time series. Hourly time step preferred. Show flood stage. Refer to flood of record and/or recent flood levels (e.g. Napa River 1986 storm of record). DHM 4 km grid too coarse; need 250 m grid for watershed. Local responders used to more fine-scale information. Need to better define discharge and runoff. CFS is not communicable – make it more relatable. | | | | | |

**Table 3.** Summary of TTE questionnaire responses for coastal flooding information.

**Tabletop Exercise Questionnaire Responses**

| Question Topic | Subtopic | % Ratings by Category | | | Weighted Score | Comments |
|---|---|---|---|---|---|---|
| | | Not Useful 1 | Some what 3 | Very Useful 5 | | |
| Please provide your first-level reaction for the information provided in the COAST section. Note that COAST data include the CoSMoS model Meteorological forcing data (winds, barometric pressure), as well as gridded model forecasts such as Waves, Currents, and Water Levels. Time series show the forecast water level heights over time at a point, so the peak height, time-to-peak and duration of high water. | Meteorological | 20% | 60% | 20% | 3.0 | Met.data rated Somewhat Useful, with range of Not Useful to Very Useful. |
| | Waves | 0% | 100% | 0% | 3.0 | Wave forecasts unanimous rated Somewhat Useful. |
| | Currents | 20% | 80% | 0% | 2.6 | Currents forecast rated Somewhat Useful; some rated Not Useful . |
| | Water Levels | 0% | 60% | 40% | 3.8 | Water level forcasts rated Somewhat Useful to Very Useful. |
| | Time Series | 0% | 40% | 60% | 4.2 | Time series forecasts rated Very Useful. |
| Additional comments on the COASTAL section. | Show tide projections. Overlay FEMA Floodplain for reference; also recent flood inundation levels. | | | | | |
| Please provide your first-level reaction for the information provided in the FLOOD INDICES section. Note that FLOOD INDICES involve CoSMoS and DHM model output data characterizing flood magnitude, extent and criticality. The Hazard Index involves time integration of currents and water depth to find the locations which will be most impacted in a storm event, either through very fast currents, extremely deep water or extremely long flood duration, or some combination of the three. | Start of Flooding | 0% | 50% | 50% | 4.0 | Flood start time rated Somewhat to Very Useful. |
| | Duration of Flooding | 0% | 0% | 100% | 5.0 | Flood duration rated unanimous Very Useful. |
| | Time of Max Depth of Flooding | 0% | 0% | 100% | 5.0 | Time of Max Depth of Flooding rated unanimous Very Useful. |
| | Max Water Depth | 0% | 25% | 75% | 4.5 | Max Water Depth rated Very Useful. |
| | Hazard Index | 0% | 50% | 50% | 4.0 | Hazard Index rated Somewhat to Very Useful. Some confusion on what it means. |
| Additional comments on the FLOOD INDICES section. | People primarily want to know - Where, How High and When. Consider describing projections as "ankle deep" "knee deep" etc. Need to consider audience – is it Emergency Managers and First Responders or someone else? Need to consider point of products – newscast or police with blowhorn saying need to evacuate? Local corroboration of projected flooding. | | | | | |
| Please provide your first-level reaction for the information provided in the IMPACTS - CRITICAL FACILITIES section. These data involve identification of built facilities that are of concern to flood responders. | Fire Stations | 0% | 75% | 25% | 3.5 | Fire stations rated Somewhat Useful. |
| | Schools | 0% | 75% | 25% | 3.5 | Schools rated Somewhat Useful. |
| | Wastewater Treatment Plants | 0% | 50% | 50% | 4.0 | Wastewater treatment rated Somewhat to Very Useful. |
| | Roads | 0% | 20% | 80% | 4.6 | Roads data rated Very Useful. |
| Additional comment(s) about the CRITICAL FACILITIES data you may use. | Suggest include hospitals and airports. Bridges subject to flooding. Identify key locations for each region...to help bring context to flood projections. EOC locations could be included in an internally (non-public) available layer - but that would not be good to include a public layer. Forecasters have hydro-database (E-19). Develop database service of user-generated content. | | | | | |

## 6. Conclusions and Next Steps

A combined watershed-coastal storm modeling system-Hydro-CoSMoS-was developed to demonstrate the feasibility of linking a distributed tributary and coastal storm model in San Francisco Bay. The demonstration was based on a direct coupling of the discharge from the watershed model into the coastal model, a first step toward a more advanced coupling planned for the AQPI system. The tributary portion of Hydro-CoSMoS generated forecast information for each grid including flow hydrographs (peak flow, time-

to-peak, duration of high flow), soil moisture, and flood recurrence level. The CoSMoS hydrodynamic model was used to calculate water levels that account for tributary inflows, tidal forcing, storm surge, and in-Bay generated wind waves derived from the wind and pressure fields of an atmospheric forecast model. The coastal component portrayed flood inundation and timing, and duration. Road-stream crossings and other critical facilities in the tributaries and coastal zone were integrated in the Hydro-CoSMoS system to help identify flood impact features in the tributaries and coastal zone.

As part of the Hydro-CoSMoS development, a storm scenario was created using data from actual events scaled up to produce a plausible combined watershed and coastal flooding event. A TTE was conducted at the culmination of the project, to engage potential stakeholders and end users and solicit feedback on the kinds of information needed from an integrated flood forecasting system in the Bay area. Key results from this study include:

- The prototype Hydro-CoSMoS was deployed and demonstrated the ability to provide watershed and coastal flood information at scales and at locations not currently served by NWS operations. The project was successful in showing how tributary flows could be used to inform the coastal storm model during a flooding scenario.
- The assessments provided an opportunity for the model developers to interact with the end-users, providing valuable information to help guide continued model development and to inform what model outputs and formats are most useful to end-users.
- Similarly, the assessments provided an opportunity for the end-users to become familiar with this emerging tool and to gain an initial level of understanding prior to advancement to real-time operations for the AQPI project. This process helps develop an engaged end-user who will be more likely to utilize the model products once it is running operationally.
- The assessments also enabled the project team to learn about other potential end-users and leverage the results of this TTE exercise to engage subsequent end-users in the San Francisco Bay region and other counties in CA.

Integrating coastal storm and fluvial events is of growing importance as a way to provide information where current operational products are not available. This study has shown that interaction of model developers with many local authorities in the design of the system is helpful in improving usefulness and allows validation in locations with little data by tapping local knowledge base and experience. Looking forward, TTE participants expressed interest in continued discussions as the Hydro-CoSMoS model continues to be built out for Napa and the other watersheds in the San Francisco Bay.

One important outcome from these interactions with users was their emphasis on integration of Hydro-CoSMoS prototype and the future AQPI system with their current flood warning procedures. Most of the counties have developed their own precipitation monitoring networks (e.g., OneRain, https://onerain.com/) which they use in conjunction with NWS precipitation forecasts. A typical application involves flood threat indices based on cumulative storm rainfall and the forecasts. They expressed great interest in higher resolution (space and time) precipitation monitoring and forecasts, above and beyond what NWS is currently able to provide, in formats that fit with their applications. Conversely, the local rainfall monitoring data has value for calibrating AQPI radar-precipitation monitoring products, as well as providing a centralized location to host all the local data and share with users. Thus, initial AQPI system development efforts are focused on establishing data exchanges to download local rainfall data to the AQPI database, and to upload AQPI precipitation products using formats compatible with the local applications. NOAA is working with other federal agencies to develop a seamless, fully coupled tributary-coastal forecast system that will eventually cover the United States. As coastal models operate best at a regional scale, it is anticipated that lessons learned from the Hydro-CoSMoS prototype in the Napa watershed and future real time deployment across all of San Francisco Bay will inform the development of the nation-wide coastal implementation.

**Author Contributions:** Conceptualization, R.C., L.E.J., L.H., J.A.F., L.E., P.B., and M.A.; methodology, L.E.J., J.K., J.A.F., R.M.-K., and L.H.; software, T.C., and G.P.; formal analysis, R.C., L.E.J., J.K.; resources, M.A.; writing—original draft preparation, L.E.J., J.K., and R.C.; writing—review and editing, L.E.J., P.B., and L.E. All authors have read and agreed to the published version of the manuscript.

**Funding:** This research was funded by a grant from the CA-DWR, grant number 2013-PSD-3BR2WWR. Support was also provided by the NOAA Physical Sciences Laboratory.

**Informed Consent Statement:** Not applicable.

**Acknowledgments:** This project was supported by a grant from the CA-DWR. The FEWS implementation of RDHM and prototyping of near real-time simulations was part of a grant provided by the NOAA Office of Oceanic and Atmospheric Research (OAR), Office of Weather and Air Quality (OWAQ), FY2015 US Weather Research Program, Hydrometeorology Testbed (HMT) and Hazardous Weather Testbed (HWT) Grant Award # NA15OAR4590151. The NOAA Physical Sciences Division provided resources to supplement project support from DWR, ancillary data sets, and access to the FEWS system for the DHM forecasts. The authors thank the three anonymous reviewers for valuable comments that improved the manuscript.

**Conflicts of Interest:** The authors declare no conflict of interest.

## Appendix A

*Tabletop Exercise Survey Results*

- Needs for Coastal Flood Forecasts:
    o NWS forecasters use radar data, flash flood guidance, rainfall reports, and river data for their warning decision making.
    o County-level staff are concerned with storm surge into flood control channels and protecting communities and ecosystem restoration projects thereby. Most counties track flood threats and coordinate warnings with Emergency Operations Centers.

- Flood Data Sources:
    o Most use NWS warnings, including from the Weather Forecast Office (WFO) and the CNRFC. Some track precipitation reports (public and private), spotter reports (phone, email, social media, and news media), USGS stream gages.
    o Some county staff have their own precipitation and stream gage networks.

- Watershed Forecasts:
    o Participants indicated that the timing needs to be linked to local time. Hourly projections at a minimum during an event are preferred.
    o The time series outputs (i.e., hydrographs) were unanimous rated Very Useful. Would like to see flood stage. Depth and velocity of flow relate to danger.
    o Local responders want to see information at a finer scale (e.g., 250 m) than currently provided by the 1-km Napa Watershed modeling.
    o Precipitation information rated Very Helpful.
    o Soil moisture rated Somewhat Helpful.
    o DHM surface runoff rated Not Useful to Somewhat Useful.
    o Recurrence interval rated Very Useful by most, but some rated Not Useful.
    o Relate water flows to citizen experience such as storms of record (preferably within a 10-year window) to allow some comprehension of projected storm event.

- Coastal Flood Forecasts:
    o Meteorological data rates Somewhat Useful. Helps establish context.
    o Wave forecasts rated unanimous Very Useful.
    o Forecast currents rated Somewhat Useful; some rated Not Useful.
    o Water level forecasts rated Somewhat Useful to Very Useful.
    o Time series forecasts rated Very Useful.

  ○  Suggest overlay FEMA Floodplain for reference; also recent flood inundation levels.

 ● Coastal Flood Indices:

  ○  People primarily want to know—Where, how high, and when.

  ○  Consider describing projections as "ankle deep" "knee deep", etc., to indicate flood stage.

  ○  Incorporate tide projections so that responders can be aware of the confluence of tides with flooding projections.

  ○  Need to consider audience—is it Emergency Managers and First Responders or someone else?

  ○  Need to consider point of products—newscasts and media or police with blow horn saying need to evacuate?

  ○  (Would be good to have) local corroboration of projected flooding.

  ○  Flood indices uniformly rated Somewhat to Very Helpful, including those for (a) Start of Flooding, (b) Duration of Flooding, (c) Time of Max Depth, (d) Max Water Depth, and (e) Hazard Index (although some confusion on what it means).

 ● Impacts—Critical Facilities:

  ○  Identification of roads and road crossings (bridges) was rated Very Useful.

  ○  Identification of fire stations, schools and wastewater treatment plants were rated Somewhat Useful. Suggest including hospitals and airports.

  ○  Identify key locations for each region to help bring context to flood projections.

  ○  Emergency Operations Center (EOC) locations could be included in an internally (non-public) available layer-but that would not be good to include a public layer.

  ○  Forecasters have hydro-database (E-19).

  ○  Develop database service of user-generated content.

  ○  Need to narrow in on the audience for each product or output. Are the outputs geared to first responders or others?

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
