# Peer review of "Assessment of Flood Forecast Products for a Coupled Tributary-Coastal Model"

_water, doi:10.3390/w13030312_

Round 1

Reviewer 1 Report

See attached file.

Reviewer 2 Report

The paper is well written and the addressed problem is valuable. I have two minor comments and believe addressing them could further improve the quality of the paper.

  1. It was not clear how the RDHM and CoSMoS are coupled. If they are not fully two-way coupled, how is the data transferred between the two software? How did the authors guarantee the convergence of the computation?
  2. As mentioned in the manuscript, RDHM is likely to be replaced by NWM in the future. Is there a plan to replace RDHM by NWM?

Reviewer 3 Report

Please see attached PDF for comments.

Round 2

Reviewer 3 Report

I appreciate the attention that the authors gave to the revision. They have thoughtfully and adequately addressed the comments from the initial review. I recommend the paper for publication in its current form.